# A Novel Feature Extraction Algorithm and System for Flexible Integrated Circuit Packaging Substrate

**DOI:** 10.3390/mi13030391

**Published:** 2022-02-28

**Authors:** Dan Huang, Juan Wang, Yong Zeng, Yongxing Yu, Yueming Hu

**Affiliations:** 1The School of Mechanical and Automotive Engineering, South China University of Technology, Guangzhou 510641, China; dan78huang@scut.edu.cn; 2The School of Automotive Science and Engineering, South China University of Technology, Guangzhou 510641, China; 201910102668@mail.scut.edu.cn (Y.Z.); permanyu@126.com (Y.Y.); auymhu@scut.edu.cn (Y.H.)

**Keywords:** flexible integrated circuit packaging substrate, level set function, defect detection

## Abstract

Aiming at the line defect detection of a flexible integrated circuit substrate (FICS) without reference template, there are some problems such as line discontinuity or inaccurate line defect location in the detection results. In order to address these problems, a line feature detection algorithm for extracting an FICS image is proposed. Firstly, FICS image acquisition is carried out by using the appearance defect intelligent detection system independently developed in our lab. Secondly, in the algorithm design of the software system, the binary image of the line image to be segmented is obtained after the color FICS image is classified by K-means, median filtering, morphological filling and closed operation. Finally, for an FICS binary image, an image segmentation model with convexity-preserving indirect regular level set is proposed, which is applied to extract the line features of an FICS image. Experiment results show that, compared with the CV model, LBF model, LCV model, LGIF model, Order-LBF model and RSF model, the proposed model can extract line features with high accuracy, and the line boundary is smooth, which lays an important foundation for high-precision measurement of line width and line distance and high-precision location of defects.

## 1. Introduction

Chip design, wafer manufacturing and packaging testing are three indispensable components in the integrated circuit industry chain. In the mid-1990s, a new type of IC packaging represented by BGA (ball grid array) and CSP (chip scale package) came out. A necessary new carrier for semiconductor chip packaging came into being, which is the IC package substrate [1]. IC packaging substrate is the key carrier in the packaging test link, which is divided into three categories: organic rigid packaging substrate, flexible packaging substrate [2] and ceramic packaging substrate [3]. The flexible integrated circuit substrate (FICS) adopts the Roll to Roll automatic processing mode of roller transmission to develop in the direction of lighter, higher integration and easier bending. As the IC industry enters the 7–14 nanometer process, compared with PCB (Printed Circuit Board) and FPC (Flexible Printed Circuit), FICS has also entered the process below 2 um, and the quality control performance requirements (such as system stability, light sources, noise and environmental pollution, etc.) for materials and manufacturing processes are also constantly improving. In the process of FICS defect detection, the traditional detection method of substrate alone cannot meet the accuracy requirements of industrial production. This further puts forward higher requirements for FICS surface defect detection accuracy technology.

At present, the non-destructive testing methods for evaluating the substrate quality are roughly divided into the following: X-ray testing [4], ultrasonic scanning testing [5], infrared thermal imaging testing [6] and surface acoustic wave testing. X-ray and infrared thermal imaging are used to detect the internal defects of the substrate. Aryan et al. [3] suggested that microelectronic surface defect detection techniques are still evolving in terms of accuracy, resolution and speed, and that the best future solution is likely to be a combination of methods such as IRT and 3D X-ray with appropriate modeling to overcome the shortcomings of each method. Ultrasonic scanning imaging and surface acoustic imaging are used to detect the appearance defects of the substrate, but the detection accuracy is more than 100 um, which is difficult to achieve high-density appearance defect detection of flexible IC packaging substrate. The characteristic images of the line, such as short-circuit and breakage defects, are obtained in image detection, and the actual line width and line distance are obtained by transitioning the image coordinate system to the world coordinate system using known quantities such as pixel distance, microscope magnification and CCD internal reference. Image defect recognition and classification is the core technology of appearance defect detection of flexible IC packaging substrate, which determines the function realization of automatic vision detection technology. The process of image processing is divided into four stages, as shown in Figure 1.

There are many kinds of circuit surface defect detection methods based on image processing [7,8,9,10,11]. The method based on image space mapping mainly reduces image noise and enhances hidden image details through mapping relationship. Tsai and Huang [12] proposed a global Fourier image reconstruction method to detect and locate small defects in aperiodic pattern images. Luo et al. [13] proposed a two-stage target detection framework based on convolutional neural network decoupling for non-significant defects in flexible packaging substrates and the similarities between various defects, in which the positioning task and the classification task pass through two specific module decoupling. Xia et al. [14] aimed at the problem of imbalance between the foreground and the background when detecting small defects in the circuit board and proposed a method for PCB based on FL-RFCN (focal loss and region-based full convolutional network) and the PHFE (parallel high-definition feature extraction) small defect detection method. Hu and Wang [15] used ResNet50 with a feature pyramid network as the backbone of feature extraction to better detect small defects on the PCB. Kim et al. [16] proposed an advanced PCB inspection system based on skip connection convolutional autoencoders for the problem of setting pass/fail criteria for small fault samples. Ding et al. [17] were inspired by the similarity measurement and proposed a multi-layer deep feature fusion method to calculate the similarity between the template and the defective circuit board.

The main contributions of this paper are: combined with the appearance defect intelligent detection system independently developed by our laboratory, aiming at the problems of line discontinuity or inaccurate line defect location in the line defect detection link of flexible circuit packaging substrate without reference template in practical engineering, a line feature detection algorithm for extracting FICS image is proposed. It lays an important foundation for high-precision measurement of line width and line distance and high-precision location of defects.

The structure of this paper is as follows: In the second part, it introduces the difficulty analysis of FICS surface defect detection and the purpose of this paper, as well as the characteristics of FICS line and the problems in the link of line defect detection in practical engineering. In the third part, the software and hardware system of appearance defect intelligent detector in our laboratory is introduced. In the fourth part, the algorithm of this paper is proposed. In the fifth part, the algorithm is applied to line extraction and verified. Finally, the full text is summarized.

## 2. FICS Surface Defect Detection and Analysis

### 2.1. Analysis of Difficulties in FICS Surface Defect Detection

FICS images are prone to the following problems in imaging, preprocessing and defect detection:1.In the imaging of an FICS image, due to the lack of hardware stability, image jitter, defocus blur and motion blur are easy to appear in image acquisition. Furthermore, it is difficult to keep the offset of the image in the horizontal and vertical directions at the micron level, which makes FICS defect detection unable to meet the requirements of high precision.2.In the preprocessing of FICS defect detection, the texture structure and noise in the image will be amplified at the same time. In the process of removing noise, it is easy to remove the original texture in the image, which brings difficulty to the subsequent defect detection.3.In the processing of FICS defect detection, when the difference between the defective region of interest and the substrate background is not strong, it increases the difficulty of substrate defect edge extraction.4.In the processing of FICS defect detection, for some line defects with large size and covering a wide circuit area, the routing path of the real substrate cannot be determined due to the lack of standard circuit routing template. It is difficult to detect the position and shape of defects.5.In FICS defect detection, the existing data structure defect features are difficult to adapt to unknown defects, that is, the generalization of feature extraction model is not strong, resulting in insufficient description of various defect features. Furthermore, it indirectly leads to the difficulty of defect target location and segmentation.

In response to the above problems, the following options are currently available for partial solutions: (1) In terms of illumination, the adverse effects on image acquisition can be minimized by selecting a light source with higher hardware power and sufficient brightness to highlight defects, so that it strictly controls the variation in ambient light sources. For example, different light source colors highlight different features, as shown in Figure 2, and different light source types bring different clarity, as shown in Figure 3. (2) In terms of hardware stability, vertical steel plates can be set up in the body and fixed to the mechanical platform. Marble can also be placed underneath the body to lower the center of gravity of the machine and improve stability. (3) In terms of real-time acquisition and clarity, the imaging system can be equipped with a high frame rate camera and a metallographic microscope. (4) Design robust and generalizable defect detection algorithms for the defect detection process.

### 2.2. The Characteristics of FICS Line Defect

In FICS defect detection, under the industrial standard, at present, the defect detection accuracy in the gold finger area of FICS is less than 2 um; for line width, line spacing and aperture, the detection accuracy is more than 2 um. Defects such as short circuit, open circuit, gap, scratch, bulge and depression are easy to occur in the production process, as shown in Figure 4. Furthermore, the generated defects affect the electrical performance of the high density FICS. In the detection process, the image segmentation technology has an important position in accurately extracting the line width and line distance of the FICS image. The line refers to the wire used to conduct current on the FICS. For wire gaps and gap-type defects, they mainly refer to open-circuit defects and short-circuit defects between line wires. Short-circuit and open-circuit defects are generally caused by the attachment of the circuit part to the flexible substrate, so that the metal wire cannot be completely separated from the carrier film.

At present, the detection efficiency of hardware and software systems in our laboratory is: the detection accuracy of line width and line distance can reach the minimum detection of 10 μm, the error is 2 μm and the detection efficiency, that is, the average processing speed, is 300 ms. The detection algorithm of line width and line distance is based on the nonreference detection method, as shown in Figure 5. The detection of line width and spacing can calculate the line width and spacing of wires, locate notch defects and short-circuit and open-circuit defects and realize the classification and location of defects. However, there are still the following problems in the detection: the line in the detection results is almost incomplete, especially at the turn of the line. In addition, there are many locations that are not defects but have defect characteristics at the same time.

## 3. Algorithm for Extracting FICS Image Line Features

The basic idea of the energy functional segmentation algorithm [18,19,20,21,22] is to transform the image segmentation problem into a mathematical problem of solving the minimum value of the function and achieve the purpose of image segmentation by controlling the evolution of the curve to the boundary. When the energy functional is a convex function, when the gradient descent method is used to solve the minimum value of the function, the unique value obtained is different from the initial condition. When the energy functional is non-convex and the gradient descent method is used to solve the minimum value of the function, the solution obtained is not unique, and different initial values may result in different local minimums [23,24,25]. In the existing segmentation models, most segmentation methods use direct regular models to avoid level set re-initialization, such as distance regular level set model (DRLSE) evolution [26]. When the energy functional reaches a minimum, the modulus of the gradient of the level set function tends to zero, that is, the level set function tends to a constant. As the number of iterations increases, the zero-level set gradually disappears, and the level setting function traverse weak edges, resulting in boundary leakage.

The purpose of this article is to extract the FICS line features with high precision. Based on the RSF [27] model, this paper combines Wu and He [28,29] to provide an image segmentation method that preserves convexity and indirect regular level sets. In this method, the convex energy functional is constructed so that the energy functional reaches the global minimum during the evolution process. This method uses the approximate weighted mean square error obtained by the local fitting function and the gray value of the inner and outer images of the contour to drive the contour to the target boundary. This method uses an auxiliary function as a regular term to weaken the excessive smoothness of the level set function, which is beneficial to improve the accuracy of FICS image line width and line distance detection.

### 3.1. Model Proposed

The total energy functional representation of the model is shown in Formula (Equation 1): (1)$$\begin{array}{c} E=E^{D}+E^{L}+E^{R}\\ =\lambda_{1}\int \left(\int K_{\sigma }\left(x-y\right)|I\left(y\right)-f_{1}\left(x\right){|}^{2}{\left(\phi (y)+1\right)}^{2}dy\right)dx\\ +\lambda_{2}\int \left(\int K_{\sigma }\left(x-y\right)|I\left(y\right)-f_{2}\left(x\right){|}^{2}{\left(\phi (y)-1\right)}^{2}dy\right)dx\\ +u\int_{\Omega }{\left(\phi \left(x\right)-\psi \left(x\right)\right)}^{2}dx+v\int_{\Omega }|\nabla \psi \left(x\right){|}^{2}dx \end{array}$$

Among them, $E^{D}$ is a data item, $E^{L}$ is a connection item and $E^{R}$ is a regular item. The specific data items are: *x* and *y* are pixel points, *y* is the pixel point in the area of *x*, I(y) is the gray value of the point *y*, $K_{\sigma }\left(x-y\right)$ is the Gaussian kernel function and $f_{1}\left(x\right)$ and $f_{2}\left(x\right)$ represent the gray approximate fitting values in different regions. The data items use $f_{1}\left(x\right)$, $f_{2}\left(x\right)$ and the gray value I(y) of the inner and outer images of the fitted contour to approximate the weighted mean square error to drive the contour to the target boundary. The Gaussian kernel function is shown in Formula (Equation 2):(2)$$\begin{array}{c} K_{\sigma }\left(u\right)=\frac{1}{{\left(2\pi \right)}^{\left(\frac{n}{2}\right)}{\left(\sigma \right)}^{n}}exp\left(-\frac{\left(|u{|}^{2}\right)}{2{\left(\sigma \right)}^{2}}\right) \end{array}$$

The Gaussian kernel function $K_{\sigma }\left(x-y\right)$ is the weight of I(y), and its local properties control the size of the fitting area. When |x−y|>3σ, $K_{\sigma }\left(x-y\right)\to 0$, the energy of the data item is only determined by the gray value of the neighborhood $\left\{y:|x-y|\leq 3\sigma \right\}$.

The total energy functional can be regarded as a function of the level set functions ϕ(x), ψ(x), $f_{1}\left(x\right)$, $f_{2}\left(x\right)$. Fix level set functions ϕ(x) and ψ(x), and minimize $f_{1}\left(x\right)$ and $f_{2}\left(x\right)$, and the result is as shown in Formula (Equation 3):(3)$$\begin{array}{c} \left\{\begin{array}{c} f_{1}\left(x\right)=\frac{K_{\sigma }\left(u\right)\left[{\left(\phi (x)+1\right)}^{2}I\left(x\right)\right]}{K_{\sigma }\left(u\right){\left(\phi (x)+1\right)}^{2}}\\ f_{2}\left(x\right)=\frac{K_{\sigma }\left(u\right)\left[{\left(\phi (x)-1\right)}^{2}I\left(x\right)\right]}{K_{\sigma }\left(u\right){\left(\phi (x)-1\right)}^{2}} \end{array}\right. \end{array}$$

Fixing ψ(x), $f_{1}\left(x\right)$, $f_{2}\left(x\right)$, and minimizing ϕ(x), the result is as shown in Formula (Equation 4): (4)$$\begin{array}{c} \left\{\begin{array}{c} \phi \left(x\right)=\frac{\left(u\psi \left(x\right)-\lambda_{1}A+\lambda_{2}B\right)}{\left(u+\lambda_{1}A+\lambda_{2}B\right)}\\ A=\int K_{\sigma }\left(x-y\right)|I\left(y\right)-f_{1}\left(x\right){|}^{2}dy\\ B=\int K_{\sigma }\left(x-y\right)|I\left(y\right)-f_{2}\left(x\right){|}^{2}dy \end{array}\right. \end{array}$$

Fixing ϕ(x), $f_{1}\left(x\right)$, $f_{2}\left(x\right)$, and minimizing ψ(x), the result is as shown in Formula (Equation 5): (5)$$\begin{array}{c} \left(-v\nabla +u)\psi (x)=u\phi (x\right) \end{array}$$

Solve by fast Fourier transform, and obtain as shown in Formula (Equation 6): (6)$$\begin{array}{c} \psi \left(x\right)=F^{-1}\left(\frac{uF(\phi (x))}{u-v(F^{*}\left(\nabla_{x}\right)F\left(\nabla_{x}\right)+F^{*}\left(\nabla_{y}\right)F\left(\nabla_{y}\right))}\right) \end{array}$$

Among them, *F* is the Fourier transform, $F^{*}$ is the conjugate of the Fourier transform and $\nabla_{x}$ and $\nabla_{y}$ represent the difference operator in the and directions, respectively. Please see Appendix A for details of the relevant proof of convexity.

### 3.2. Algorithm Design

Based on the above line features of an FICS image, the segmentation model proposed in this paper is only suitable for target segmentation and edge extraction of a binary image. The main innovation of this paper is that, in the fourth step of the proposed algorithm, based on the RSF [27] model, this paper combines Wu and He [28,29] to provide an image segmentation method that preserves convexity and indirect regular level sets in order to extract FICS line features with high accuracy.

The algorithm for extracting FICS line features in this paper is shown in Figure 6. The details are as follows: the first step is to input the color ultra-thin and high-density FICS image; in the second step, the color ultra-thin and high-density FICS images are classified by K-mean, and the number of clusters is 2; in the third step, median filtering, morphological filling and closing operations are carried out, respectively, to obtain the binary image of the line image to be segmented; and in the fourth step, the convex preserving indirect regularization model proposed in this paper is used to extract the edge of image lines.

The design reasons of this algorithm are as follows: This paper selects K-means algorithm for image binary processing to obtain the binary image. Its purpose is to save the time of the whole algorithm by eliminating the problem of threshold selection from gray image to binary image. Compared with the mean filter, the median filter is better at dealing with impulse noise, such as salt and pepper noise, and it is easier to protect the sharp corners or edges of the image. The filling hole algorithm is used for morphological filling in order to highlight the boundary of the target area and remove the undeleted outliers in the line. The purpose of this paper is to highlight the boundary and corner of the target area and pave the way for accurate extraction of FICS line features. The purpose of applying this model to the last step of this algorithm is to segment the binary image using this model to extract FICS line features. This process has a global minimum, which is beneficial to improve the accuracy of extracting line features.

## 4. Intelligent Detection System

### 4.1. Hardware System Structure

Figure 7 shows the physical image of the intelligent detector for the appearance defect of the flexible IC package substrate. The hardware core is composed of X, Y, Z three-axis motion module frame equipped with high and low imaging systems. The three-axis motion module is mainly responsible for the movement of the imaging system on the working platform. The imaging system is mainly responsible for the image acquisition of the area to be inspected. A sophisticated motion platform is the key to accurate imaging of the imaging system. The vision imaging system is mainly composed of industrial cameras, industrial lenses and machine vision light sources. When the area to be inspected is too large and exceeds the single field of view of the camera, it is necessary to control the precision motion platform equipped with the camera by the industrial computer for image acquisition. This article uses a three-axis motion module to achieve large-scale image acquisition. The platform includes key accessories such as motors and drivers, ball screws, stages, limit sensors and drive boards.

### 4.2. The Design and Application of Software System

The key components of the software are mainly divided into seven units: etching detection unit, drilling detection unit, appearance detection unit, motion control unit, data storage unit, intelligent analysis unit and view unit, as shown in Figure 8. The functions of etching process, drilling process and appearance process are similar, but they mainly face different application objects and different detection accuracy.

The etching process mainly detects the size (mm) of 25 × 25 470 × 470, and the thickness (mm) of the line width/line distance, open/short circuit, oxidation, foreign matter, air bubble, golden finger, etc., of the pattern of 0.5–3.0. The detection accuracy can reach 70 nm;The drilling process mainly detects round holes, voids, pinholes and other types of defects with a size (mm) of 25 × 25 470 × 470 and a thickness (mm) of 0.5 to 3.0. The detection accuracy ranges from 0.5 μm to 10 μm;The appearance process mainly detects multi-scale appearance defects such as open circuit/short circuit, indentation, oxidation, bumps, scratches/scratches, wrinkles/wrinkles, foreign matter, bubbles, appearance color, ink, stains, cracks, warpage, etc. [7,30,31]. The detection accuracy can reach 2 μm to 10 μm;The data storage unit is mainly responsible for real-time storage of defect information during the detection process and can provide functions such as querying defect records and deleting information in other states. It can also provide data support for the intelligent analysis unit;The motion control unit realizes the control command transmission and message feedback between the industrial control host and the motion control board through serial communication;The intelligent analysis unit can perform statistics and monitoring of key physical parameters such as line width/line distance, circularity and aperture of flexible IC packaging substrates and conduct stability evaluation of the manufacturing process;The view unit is mainly a visual display of various types of information.

The task flow chart of the appearance defect intelligent detector is shown in Figure 9. After the detection task starting, the motor needs to be driven to move and collect pictures. At this time, there are two ways to obtain pictures: one is through a common zoom lens system, and the other is through a metallurgical microscope. After the pictures are acquired, different detection algorithms are required for different processes. The types can be roughly divided into three categories: appearance defect detection algorithms, drilling processes and etching processes.

## 5. Experiment of Extracting Line Features from FICS Image

### 5.1. Evaluating Indicator

In order to measure the effect of image segmentation, the segmented image can be evaluated quantitatively. Firstly, three evaluation methods based on pixel are used, namely accuracy (Ac), sensitivity (Se), specificity (Sp). The three indicators are defined as follows:(7)$$\begin{array}{c} Ac=\frac{TP+TN}{TP+FP+TN+FN} \end{array}$$
(8)$$\begin{array}{c} Se=\frac{TP}{TP+FN} \end{array}$$
(9)$$\begin{array}{c} Sp=\frac{TN}{FP+TN} \end{array}$$

In Formulas (Equation 7)–(Equation 9), TP represents the number of correctly divided target pixels, FP represents the number of pixels that belong to the background but are wrongly divided into the target, TN represents the number of correctly divided background pixels and FN represents the number of pixels that belong to the target but are wrongly divided into the background. Accuracy (Ac) represents the probability that the correctly segmented defect region and background region account for all regions of the image. The greater the sensitivity (Se), the specificity (Sp), the lower the probability of misjudging.

The other two objective evaluation indexes, Dice similarity coefficient (DSC) [32] and Jaccard similarity (JS) [33], are defined as follows:(10)$$\begin{array}{c} DSC=\frac{2N\left(S_{1}\cap S_{2}\right)}{N\left(S_{1}\right)+N\left(S_{2}\right)} \end{array}$$
(11)$$\begin{array}{c} JS=\frac{N\left(S_{1}\cap S_{2}\right)}{N\left(S_{1}\cup S_{2}\right)} \end{array}$$

In Formulas (Equation 10) and (Equation 11), $S_{1}$ and $S_{2}$ respectively represent the results obtained by various segmentation methods and the results obtained by expert manual segmentation. For the standard segmented image of metallographic image surface distribution, according to the on-site process requirements, experts judge the surface distribution by observing the color depth and particle morphology of metallographic image surface components under different light sources and then obtain the standard segmented image. $N\left(\cdot \right)$ represents the area in the segmentation result. DSC indicates the proximity between the segmentation result and the real contour. JS is used to compare the similarity and difference between limited sample sets. The value range of DSC and JS is between 0 and 1. The larger their value, the sample similarity is higher, and vice versa.

Mutual information (MI) is an important concept in information theory which describes the correlation between two systems or the amount of information contained in each other. In image registration, the mutual information of two images reflects the mutual inclusion degree of information between them through their information entropy and joint entropy. When the similarity of two images is higher or the overlapping part is greater, the correlation is greater, and the joint entropy is smaller, that is, the mutual information is greater. For images *R* and *F*, their mutual information is expressed as:(12)$$\begin{array}{c} MI\left(R,F\right)=H\left(R\right)+H\left(F\right)-H\left(R,F\right) \end{array}$$

Among them, H(R) and H(F) are information entropy, which are used to describe the measure of system uncertainty and reflect the total amount of information provided by a system itself. H(R,F) is the joint entropy. For an image, the calculation expression of its information entropy is as follows: (13)$$\begin{array}{c} p_{i}=\frac{h_{i}}{\sum_{i=1}^{N-1}h_{i}} \end{array}$$
(14)$$\begin{array}{c} H\left(Y\right)=-\sum_{i=0}^{N-1}p_{i}\log p_{i} \end{array}$$

$h_{i}$ represents the total number of pixels with a gray value of *Y* in the image *i*. *N* represents the gray level of the image *Y*. $p_{i}$ represents the probability of occurrence of gray *i*.

When the two images have similar gray distribution, mutual information is prone to false matching. Mutual information is sensitive to the overlapping area between two images. If the overlapping area is too small, the mutual information will be very small, and the registration accuracy will be reduced. Normalized mutual information (NMI) is more robust than mutual information evaluation index, which is defined as follows: (15)$$\begin{array}{c} NMI\left(R,F\right)=\frac{H(R)+H(F)}{H(R,F)} \end{array}$$

### 5.2. Experimental Analysis

In the precision electronic defect detection, the hardware system and software system of FICS image acquisition are in the third part, which will not be repeated here. In order to verify the effectiveness and rationality of the algorithm in this paper, MATLAB r2017a is used for simulation. We extracted three main types of line features: turning areas of the line, short circuits and broken circuits based on the problems that arise in real line detection as shown in Figure 5. There are 100 image samples of each type of feature, each containing only one feature.

Firstly, the experiment is carried out on the conductor at the turn of the normal line. Set initialization parameters: space step is 1, time step is 0.1, Gaussian kernel variance σ=20, $\lambda_{1}=1$, $\lambda_{2}=1$, u=1, v=1. Initialize the level set function ϕ(x) and auxiliary function ψ(x) to make $\phi^{0}=\psi^{0}$; the number of iterations is 40, and the evolution is stopped. As shown in Figure 10, Figure 10a is the original image, Figure 10b is the image classified by the K-means algorithm, Figure 10c is the image after median filtering, Figure 10d is the image after the morphological filling hole algorithm, Figure 10e is the image after the closing operation and Figure 10f is the image obtained after the feature extraction of the model in this paper. The level set function after evolution is shown in the left picture of Figure 11 and the pixel coordinates extracted from the edge of the line are shown in the right picture of Figure 11.

Secondly, experiments are conducted on the lines with short-circuit and open-circuit defects. Set initialization parameters: space step is 1, time step is 0.1, Gaussian kernel variance σ=20, $\lambda_{1}=1$, $\lambda_{2}=1$, u=1, v=1. Initialize the level set function ϕ(x) and auxiliary function ψ(x) to make $\phi^{0}=\psi^{0}$. For short-circuited display images, the number of iterations is 40 times and the evolution is stopped as shown in Figure 12, and the results of the evolution of the level set function are shown in Figure 13; for the broken display images, the number of iterations is 300 times and the evolution is stopped as shown in Figure 14, and the results of the evolution of the level set function are shown in Figure 15. Through experimental observation, firstly, Figure 10b, Figure 12b and Figure 14b are a binary image obtained by the K-means binary classification algorithm, but these images are accompanied by a lot of noise; Figure 10c–e show that image filtering can be achieved and line features can be highlighted; Figure 10f, Figure 12f and Figure 14f show that the line features can be extracted accurately, smoothly and continuously no matter if at the line defect or the turning point in the normal line. It is an important pavement for high-precision measurement of line width and line distance and high-precision positioning of defect locations. Secondly, the left images in Figure 11, Figure 13 and Figure 15 show that the evolution function after iteration has a global minimum, and the image coordinates are shown on the right. It proves the validity and rationality of the algorithm proposed in this paper.

In order to verify the effectiveness of the method proposed in this paper, the following comparison experiments of different segmentation models are carried out. Compare the model proposed in this paper with the CV model [34], LBF model [35], LCV model [36], LGIF model [37], Order-LBF model [38] and RSF model. In each method, the initial contour position of the level set is set to be the same, and the number of iterations of the level set evolution is 40 and the remaining relevant parameters are consistent with the original reference. Table 1 shows the evaluation indexes of the segmented images obtained after different segmentation models are used in the feature extraction of short-circuit lines.

By analyzing Table 1, it can be obtained that both the LBF model and the Order-LBF model have low accuracy, sensitivity, specificity, Dice similarity coefficient, Jaccard similarity, mutual information and regular mutual information evaluation indicators; the CV model has the highest specificity evaluation index. It has the highest accuracy, sensitivity, Dice similarity coefficient, Jaccard similarity, mutual information and canonical mutual information evaluation index after extracting line features through the method of this paper. To further illustrate, the LBF model and Order-LBF model cannot effectively extract FICS line features; the CV model can more effectively minimize the probability that the characteristic part is misjudged as the background. The method proposed in this paper can extract FICS line features more effectively and accurately, which once again confirms the effectiveness and rationality of the algorithm proposed in this paper.

## 6. Conclusions

This paper proposes a feature detection algorithm for extracting images to extract FICS line turn areas, short circuits and break features based on the problem of line discontinuities or inaccurate line defect locations in practical engineering. In the final part of this algorithm, an image segmentation of indirect rule level set with convexity preservation is proposed for FICS binary images. Experimental results show that the proposed model can extract line features more accurately than other variational segmentation models, and the line boundaries are smooth, laying an important foundation for measuring line width and line distance with high accuracy and locating defects with high precision.

However, the algorithm proposed in this paper contains a variational level set segmentation model. When the level set function is updated in one iteration, it needs to be computed again for all pixel points of the image, resulting in a boost in computational effort and making the algorithm run longer. In our experiments, the line feature extraction time for each FICS exceeds 100 s, which does not meet the real-time requirements in real-world engineering. How to ensure that the model in this chapter reduces the computation time while the segmentation accuracy remains the same is the next research focus.

## Figures and Tables

**Figure 1 micromachines-13-00391-f001:**
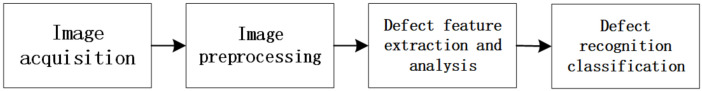
Inspection and treatment process of appearance defects.

**Figure 2 micromachines-13-00391-f002:**
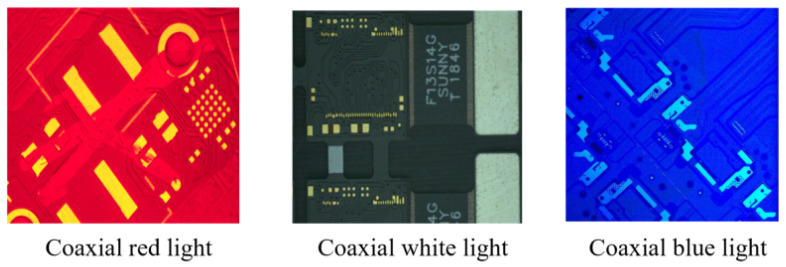
Comparison of the collection effect of different colors of light under the coaxial light source.

**Figure 3 micromachines-13-00391-f003:**
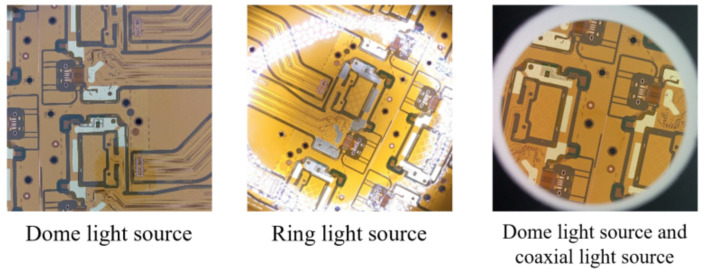
Comparison of the collection effect of the same type of substrate under different types of light sources.

**Figure 4 micromachines-13-00391-f004:**
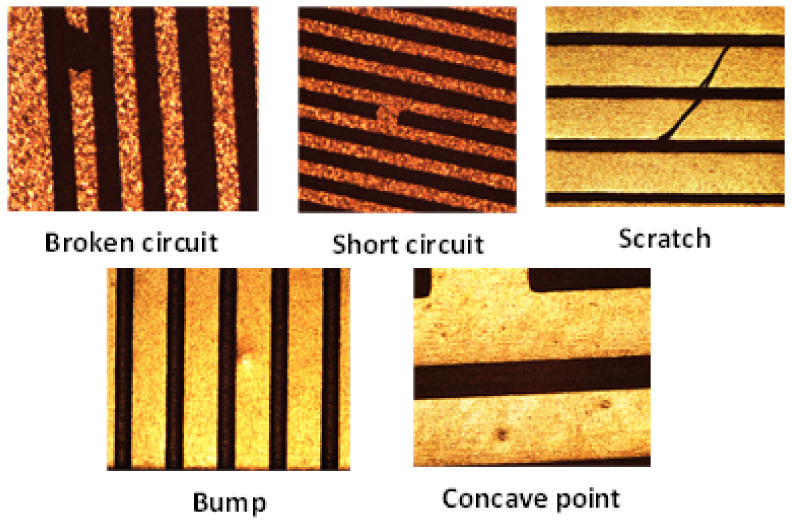
FICS line defects.

**Figure 5 micromachines-13-00391-f005:**
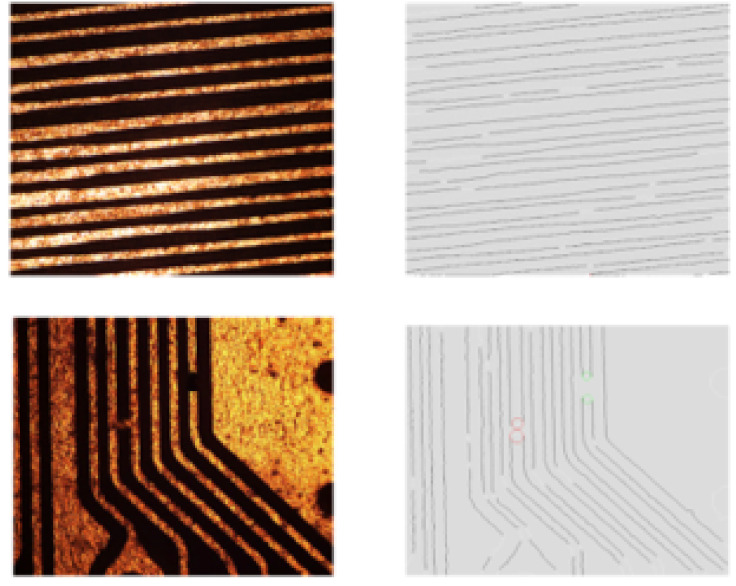
Line test results in actual engineering.

**Figure 6 micromachines-13-00391-f006:**
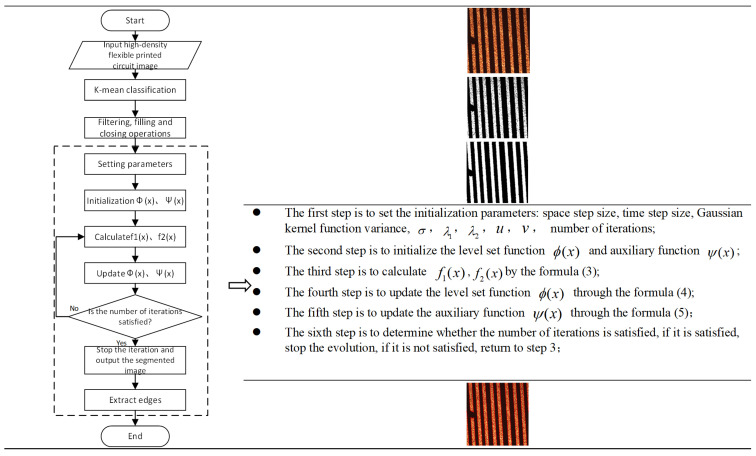
Schematic diagram of the main interface of the system.

**Figure 7 micromachines-13-00391-f007:**
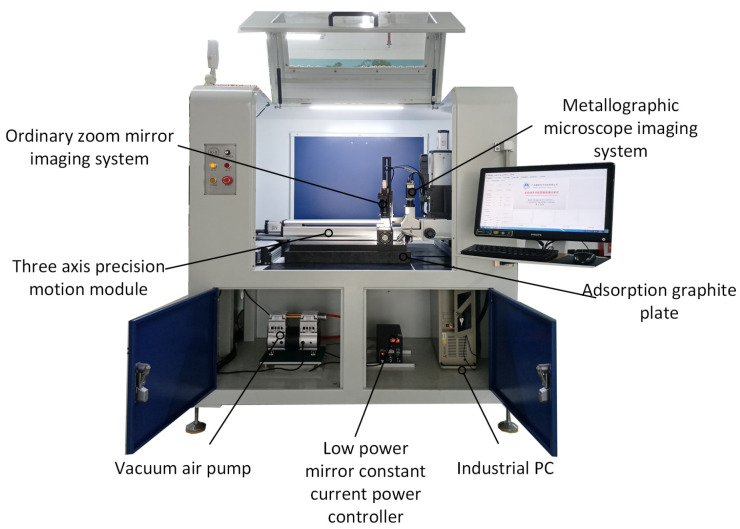
Physical image of the intelligent detector for the appearance defect of FICS.

**Figure 8 micromachines-13-00391-f008:**
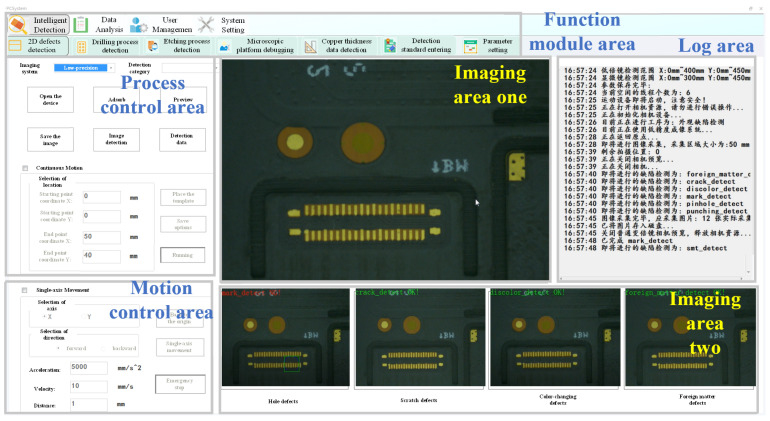
Schematic diagram of the main interface of the system.

**Figure 9 micromachines-13-00391-f009:**
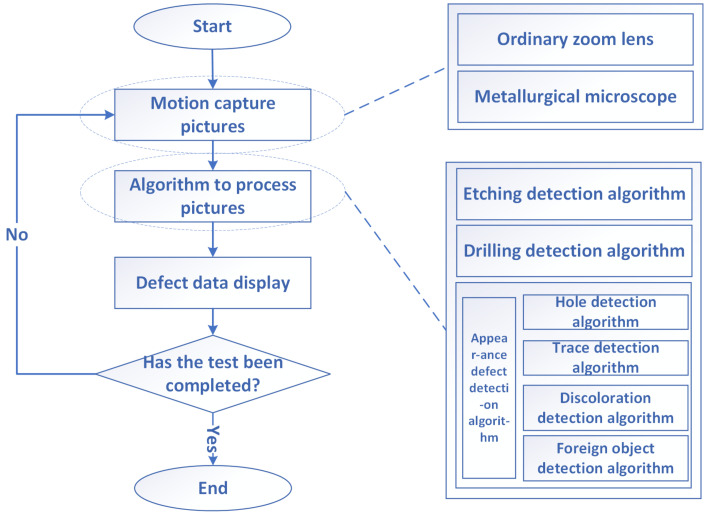
Schematic diagram of the main interface of the system.

**Figure 10 micromachines-13-00391-f010:**
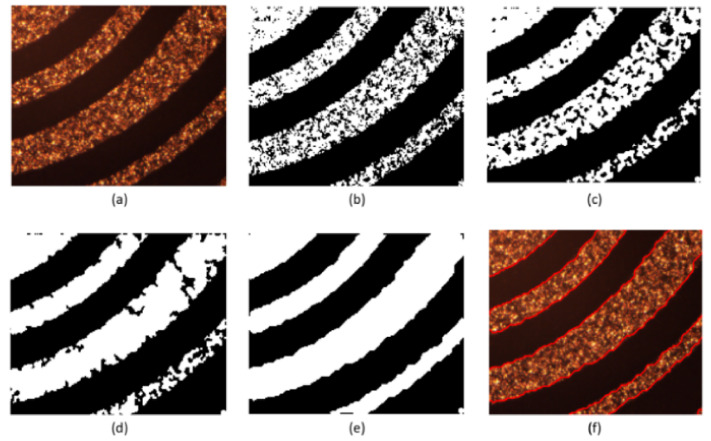
Display diagram of the algorithm for feature extraction of normal lines. (**a**) is the original image; (**b**) is the image classified by the K-means algorithm; (**c**) is the image after median filtering; (**d**) is the image after the morphological filling hole algorithm; (**e**) is the image after the closing operation and (**f**) is the image obtained after the feature extraction of the model in this paper.

**Figure 11 micromachines-13-00391-f011:**
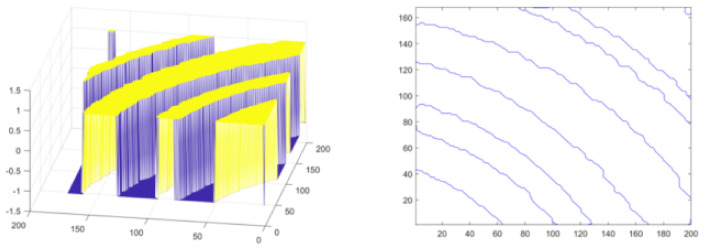
Evolution function and line extraction diagram after iteration.

**Figure 12 micromachines-13-00391-f012:**
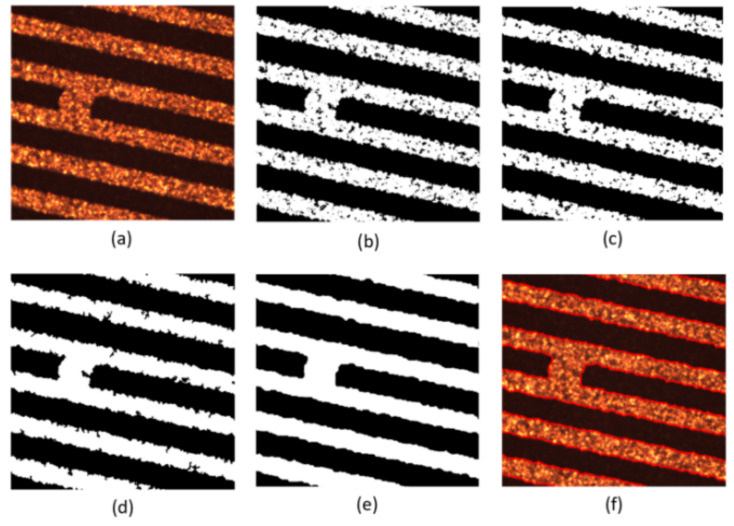
Display diagram of the algorithm for short-circuit defect feature extraction. (**a**) is the original image; (**b**) is the image classified by the K-means algorithm; (**c**) is the image after median filtering; (**d**) is the image after the morphological filling hole algorithm; (**e**) is the image after the closing operation and (**f**) is the image obtained after the feature extraction of the model in this paper.

**Figure 13 micromachines-13-00391-f013:**
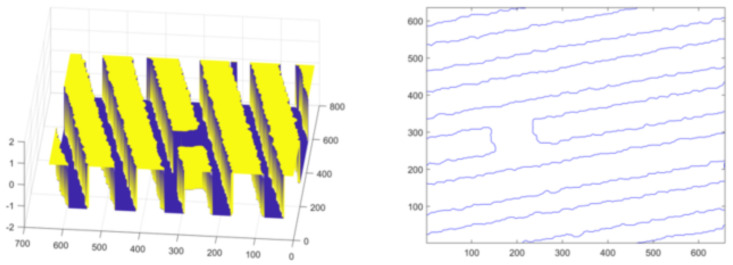
Evolution function and line extraction diagram after iteration.

**Figure 14 micromachines-13-00391-f014:**
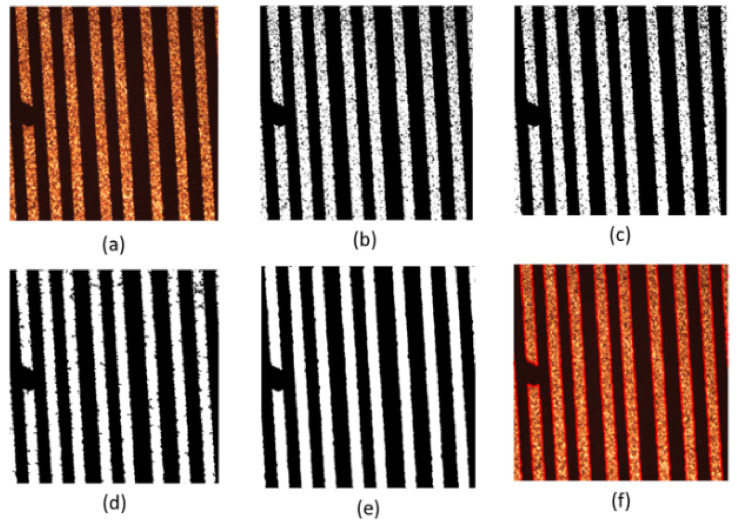
Display diagram of the algorithm for feature extraction of open defects. (**a**) is the original image; (**b**) is the image classified by the K-means algorithm; (**c**) is the image after median filtering; (**d**) is the image after the morphological filling hole algorithm; (**e**) is the image after the closing operation and (**f**) is the image obtained after the feature extraction of the model in this paper.

**Figure 15 micromachines-13-00391-f015:**
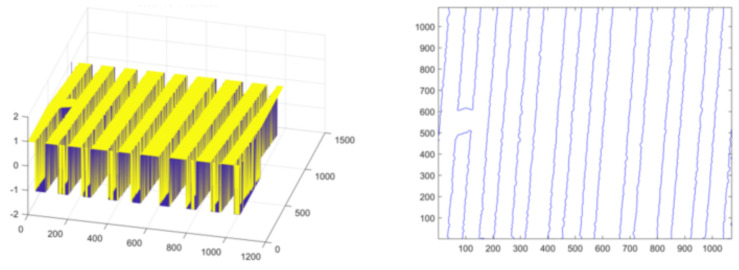
Evolution function and line extraction diagram after iteration.

**Table 1 micromachines-13-00391-t001:** Evaluation index of short-circuit line feature extraction based on different segmentation models.

Index	Ac	DSC	JS	MI	NMI	Se	Sp
CV model	0.8939	0.9086	0.8324	0.5701	1.4139	0.7797	0.9955
LBF	0.5918	0.6864	0.5225	0.0238	1.0136	0.3084	0.8438
LCV	0.9077	0.9195	0.8509	0.6068	1.4491	0.8090	0.9954
LGIF	0.7686	0.8198	0.6946	0.3068	1.2053	0.5145	0.9944
Order-LBF	0.5777	0.5880	0.4164	0.0176	1.0089	0.5870	0.5694
RSF	0.7037	0.7574	0.6095	0.1308	1.0744	0.5123	0.8738
OURS	0.9657	0.9684	0.9386	0.7934	1.6636	0.9370	0.9912

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
