# Peer review of "A Novel Feature Extraction Algorithm and System for Flexible Integrated Circuit Packaging Substrate"

_micromachines, 2022, doi:10.3390/mi13030391_

Round 1

Reviewer 1 Report

The paper presents the hardware structure and software design of the appearance defect intelligent detection system, and mainly uses it to obtain FICS (flexible integrated circuit packaging substrate) images and takes them as the research object. In the process of FISC surface defect line detection, for the problems of discontinuous line detection and inaccurate defect location, this paper mainly proposes a convexity-preserving indirect level set model. The proposed model is combined with traditional methods (K-means, median filtering, morphological filling and closed operation) to extract FICS line features. Compared with other models, the proposed scheme obtains higher accuracy of line features.

The paper is generally clear, and its theoretical scheme is scientific and reasonable, the experimental design is suitable for the verification of the proposed method. Most of the references cited are within the last 5 years, and the paper doesn’t include an abnormal number of self-citations.

Some suggestions for amendments to the paper as follow:

    • The paper focuses on discussing a method of FICS line feature extraction. From the perspective of logic and rationality, I think it should first discuss the extraction method, and then introduce the intelligent detection system (it is used to verify the relevant methods). Therefore, in terms of the structure of the article, it is recommended to adjust the order of the third part 'Intelligent Detection System' and the fourth part 'Algorithm for Extracting FICS Image Line Features'.
    • In Subsection 5.1., the paper used too long to introduce evaluation metrics. It is recommended to simplify this section.
    • In Figure 9, Figure 11, and Figure 13, please use English uniformly for the labels.
    • Please check the text carefully for details, such as no punctuation on lines 224 and 234.

Author Response

  1. The paper focuses on discussing a method of FICS line feature extraction. From the perspective of logic and rationality, I think it should first discuss the extraction method, and then introduce the intelligent detection system (it is used to verify the relevant methods). Therefore, in terms of the structure of the article, it is recommended to adjust the order of the third part 'Intelligent Detection System' and the fourth part 'Algorithm for Extracting FICS Image Line Features'.

Response: Thank you very much for your professional comment. We have adjusted the structure of this article, and marked the subtitle in red.

  1. In Subsection 5.1., the paper used too long to introduce evaluation metrics. It is recommended to simplify this section.

Response: Thank you very much for your professional comment. We have simplified this part, and marked the change part in red.

  1. In Figure 9, Figure 11, and Figure 13, please use English uniformly for the labels.

Response: Thank you very much for your professional comment. We have modified the labels, and marked it in red

  1. Please check the text carefully for details, such as no punctuation on lines 224 and 234.

Response: Thank you very much for your professional comment. We have checked the whole text and marked the change part in red.

Reviewer 3 Report

The results shows good efficiency indexes. However, the computing costs such as time, complexity are not discussed. In addition, please also suggest theoretical issues that make your method better in efficiency.

Author Response

The results shows good efficiency indexes. However, the computing costs such as time, complexity are not discussed. In addition, please also suggest theoretical issues that make your method better in efficiency.

Response: Thank you very much for your reasonable suggestion and professional comment. The proposed algorithm in this paper contains a variational level set segmentation model. When the level set function is updated in one iteration, for all the pixel points of the image, it needs to be computed again, resulting in a boost in computation and making the algorithm run longer. In our experiments, the line feature extraction time for each FICS exceeds 100s, which cannot meet the real-time requirement in practical engineering. Therefore, we do not consider it necessary to list the computing costs. How to ensure that the model in this chapter reduces the computation time while the segmentation accuracy remains the same is the next research focus.

We have summarized the above contents in the conclusion section of the article and marked it in red.

Round 2

Reviewer 2 Report

Dear Authors, 

      Thanks for your efforts to improve the paper. There is only 2 comments that I hope the authors to help: Figure 8 there are some text in Chinese. I can understand that perhaps the user interface is built in Chinese. But, will it possible to explain these log-message in the text for the non-Chinese readers? Another, the resolution of Figures 8, 13 and 15 seems not enough. Can you be so kind to improve? Thanks. 

Best Regards, 

Reviewer